# The Influence of Background Ultrasonic Field on the Strength of Adhesive Zones under Dynamic Impact Loads

**DOI:** 10.3390/ma14123188

**Published:** 2021-06-09

**Authors:** Grigory Volkov, Andrey Logachev, Nikolai Granichin, Ya-Pu Zhao, Yin Zhang, Yuri Petrov

**Affiliations:** 1Mathematical and Mechanical Faculty, Saint Petersburg University, Universitetskiy Prospect 28, 198504 Saint Petersburg, Russia; andre.log@bk.ru (A.L.); kolya30.87@yandex.ru (N.G.); y.v.petorv@spbu.ru (Y.P.); 2Institute of Problems in Mechanical Engineering of the Russian Academy of Sciences, V.O. Bolshoy pr-t 61, 199178 Saint Petersburg, Russia; 3School of Engineering Science, University of Chinese Academy of Sciences, 19A Yuquanlu, Beijing 100049, China; yzhao@imech.ac.cn (Y.-P.Z.); zhangyin@lnm.imech.ac.cn (Y.Z.); 4State Key Laboratory of Nonlinear Mechanics (LNM), Institute of Mechanics, Chinese Academy of Sciences, Beisihuangxilu 15, Beijing 100190, China

**Keywords:** dynamic impact, background ultrasonic field, adhesive joint strength, incubation time criterion, fracture dynamics

## Abstract

The influence of background ultrasonic field on the ultimate dynamic strength of adhesive joints is studied using fracture mechanics analysis. Winkler foundation-type models are applied to describe the cohesion zone, and the incubation time fracture criterion is used. The challenging task is to study whether relatively weak ultrasound is able to decrease the threshold values of the external impact load depending on a joint model, such as an “elastic membrane” or “beam” approximation, and various boundary conditions at the ends. The specific task was to investigate the case of short pulse loading through application of time-dependent fracture criterion instead of the conventional principle of critical stress. Three different load cases, namely, step constant force, dynamic pulse, and their combination with ultrasonic vibrations, were also studied. The analytical solution to the problem demonstrates that background vibrations at certain frequencies can significantly decrease threshold values of fracture impact load. Specific calculations indicate that even a weak background sonic field is enough to cause a significant reduction in the threshold amplitude of a dynamic short pulse load. Additionally, non-monotonic dependency of threshold amplitude on pulse duration for weak background field was observed, which demonstrates the existence of optimal regimes of impact energy input. Moreover, this phenomenon does not depend on the way in which the beam edges mount, whether they are clamped or hinged, and it could be applied for micro-electro-mechanical switch design processes as an additional tool to control operational regimes.

## 1. Introduction

Mechanical adhesion models are extensively studied in various mechanical systems with the aim of estimating the strength of a joint. There have been many attempts at developing a mechanical model simulating the gecko’s unique ability to control the cohesive strength of its toes [1,2,3,4]. The frictional adhesion model was suggested to explain the very low detachment forces observed [5], and an artificial material with gecko-like adhesive properties was designed [6]. The gecko’s unique ability to move its feet quickly means that it is able to control adhesion force. One of the possible explanations for this could be the presence of a temporary vibrational background field that is somehow initiated by the gecko. This phenomenon was experimentally studied in [7], and it was demonstrated that the value of pull-off force depends on background vibration parameters. This result emphasizes the importance of this paper’s task to study the influence of an ultrasonic field on the strength of adhesive zones. Another important application of adhesive models is in microelectromechanical systems (MEMS), where stiction is an essential part of a joint mechanism. The basic operating principle of a typical nanoelectromechanical switch consists of deflecting an active element into physical contact with an electrode using electrostatic forces [8,9]. A closed switch contains an adhesive joint in which electrostatic force and van der Waals forces maintain contact against elastic residual stress. As the circuit voltage lowers, the electrostatic force decreases, and residual stress is able to produce an interfacial failure in the volatile switch mode. The non-volatile mode corresponds to cases in which the adhesive force exceeds that of the return elastic stress and the circuit stays closed with zero voltage and no current. Conventional means to avoid the non-volatile mode utilize either a distance increase between the active element and the electrode or a change in the adhesion surface properties of the electrode. Common methods to modify adhesion properties include surface coatings of different materials and making dimples on the electrode. The main disadvantage of surface coating is a growth in pull-in voltage level and breakdown current that can lead to the ablation of the active element or the electrode. Moreover, there is an increase in the beam bending amplitude with a subsequent intensification of the fatigue fracture process. Furthermore, dimples can produce an unstable contact which requires additional technological operations to make the switch.

Methods to achieve a volatile mode of switch mentioned thus far rely on an alteration of physical or dimensional parameters of the system. A further aim of the present study is to investigate another possible mechanical technique for switch-mode control. A vibrational background field with relatively weak intensity may be an effective tool for altering MEMS adhesion properties. Previous studies have shown the significant influence of ultrasonic vibration on a phase state of condensed matter [10,11] or energy consumption in ultrasonic-assisted machining processes [12,13]. In some cases, additional vibration that is negligible in comparison to the main process forces may also reduce their values. It was shown that this phenomenon was related not only to resonance but also to the change in load type from quasi-static to dynamic. The influence of background vibrations was demonstrated when the adhesive joint was considered to be a string on the elastic foundation [14]. For particular frequencies, oscillations of the background field with an order of amplitude lower than that of the rupture force magnitude without background vibrations dramatically decreased this threshold value.

A Winkler foundation is one of the simplest mechanical models of the adhesion zone [15], and it was chosen as the base tool for analysis in the present research. This approach remains popular despite the fact that it was suggested one and a half centuries ago. There are now many extended Winkler models in which various rheological measurements of the beam foundation are considered [16,17,18]. Beam bending under interaction of Winkler’s elastic foundation has even been implemented into the finite element method [19], and the numerical simulation is also a common way to study dynamic problems of adhesive strength analytically. It enables investigation of complicated tasks, such as the evaluation of the compressive strength of composite laminates under transverse [20] or longitudinal [21,22] dynamic impacts. It should be noted that, generally, numerical methods are not able to predict new phenomena since they provide results for certain values of initial parameters and do not indicate their influence on the final result. Thus, the simplest one-dimensional model of a beam and the two-dimensional model of a membrane on the elastic foundation are analyzed in the present study. The influence of all possible boundary conditions, including hinged, clamped, and free ends, on the strength of adhesive joints is also considered. Analytical solutions of the stated problems provide the temporal dependence of an upper layer deflection for the chosen point. In the analysis, a peeling load is applied to the center point of the beam/membrane. It is assumed that the adhesion joint fracture occurs when the elastic link of the loading point breaks, which is governed by the incubation time-critical condition [23].

This article has the following structure: Section 2 describes incubation time criterion concepts in relation to the delamination problem. Section 3 and Section 4 are devoted to the analytical solution to the problem in membrane and beam approximations of the adhesive zone. Results and discussions are covered in Section 5, and Section 6 describes some of our conclusions.

## 2. Delamination Criterion

The incubation time fracture criterion was also used as the delamination condition for the link. The application of this criterion is stipulated by its ability to predict the point of fracture in a wide range of strain rates and pulses over various time profiles [23]. The criterion is normally written in terms of stresses, whereas, here, the governing differential equations (GDEs) of the problem were formulated in terms of displacements. Therefore, the assumption about the elastic behavior of the link must be considered because it allows the criterion to be re-written using the displacement function [14]:(1)$$\underset{t\in \left[0,t_{c}\right]}{\max}\frac{1}{\tau }\overset{t}{\underset{t-\tau }{\int }}u\left(\eta \right)d\eta \leq u_{c},$$
where u(t) is the temporal dependence of a link point displacement, $u_{c}$ is a limiting elongation of the link under quasistatic loading, τ is the fracture incubation time characterizing link dynamic strength and considered to be a material property, and $t_{c}$ is the upper border of the considered time period. It also corresponds to the conventional criterion of critical stress/elongation in the case of slow static loading. If there is only one moment of time $t\in \left[0,t_{c}\right]$ turning the criterion to equality, then a threshold load fracture scenario is realized. Thus, all the threshold amplitudes of external loads are calculated according to the incubation time approach, which is known to be an effective tool to predict the threshold characteristics of fracture in both homogenous and non-homogenous materials [24,25,26]. It should also be noted that the incubation time criterion was successfully applied to the similar problem of dynamic fracture propagation in a discrete chain, and it explained the influence of a fracture criterion on admissible regimes [27].

## 3. Membrane Approximation

This section includes the simulation of the adhesive zone as a circular membrane on an elastic base. In line with the assumption outlined above, the membrane resists only to tensile forces and not to bending or torsion. A schematic diagram of the system under consideration is shown on Figure 1. Accordingly, the GDE to describe the system behavior is as follows:(2)$$\frac{\partial^{2}u\;}{\partial x^{2}}-\frac{1}{c^{2}}\frac{\partial^{2}u\;}{\partial t^{2}}-\frac{\omega^{2}}{c^{2}}u=-f\left(x,t\right),$$
where $x=\left(x_{1},\;x_{2}\right)$, u(x,t) is the displacement of the membrane from the initial zero position (m), c is the wave velocity $\left(\frac{m}{s}\right)$, ω is the characteristic of the elastic foundation of rigidity ($s^{-1}$), f(x,t) is the external peel force ($m^{-1}$), and R is the radius of the circular membrane (m). The term $\omega^{2}/c^{2}u$ describes the effect of the elastic base on the membrane.

Polar coordinates are more convenient for the analysis due to the axial symmetry of the problem. The introduction of radial coordinate r leads to the following form of the GDE:(3)$$\frac{\partial^{2}u\;}{\partial r^{2}}+\frac{1}{r}\frac{\partial u}{\partial r}-\frac{1}{c^{2}}\frac{\partial^{2}u\;}{\partial t^{2}}-\frac{\omega^{2}}{c^{2}}u=-f\left(r,t\right),\;r\in \left(0,R\right),\;\;\;\;\;t>0\;.$$The following boundary and initial conditions correspond to rigid fixation of the membrane edges and its initial zero position:(4)$$u\left(r,0\right)=\frac{\partial u}{\partial t}\left(r,0\right)=0\;u\left(R,t\right)=0.$$The dimensionless variables give a new form of the GDE that is more suitable for numerical analysis:(5)$$r^{\prime }=\frac{r}{R},\;t^{\prime }=\frac{t}{\tau },\;\omega^{\prime }=\tau \times \omega ,\;c^{\prime }=\frac{\tau }{R}\times c,$$
where τ is the incubation time of the fracture. In the following analysis, the primes in the notations are omitted for convenience. Thus, the original problem (3) and (4) gains the following dimensionless form of the GDE and initial and boundary conditions:(6)$$\frac{\partial^{2}u\;}{\partial r^{2}}+\frac{1}{r}\frac{\partial u}{\partial r}-\frac{1}{c^{2}}\frac{\partial^{2}u\;}{\partial t^{2}}-\frac{\omega^{2}}{c^{2}}u=-R^{2}\times f\left(r,t\right),$$
(7)$$u\left(r,0\right)=\frac{\partial u}{\partial t}\left(r,0\right)=0,\;u\left(R,t\right)=0.$$The solution to problem (6) and (7) is shown in Section A.1 and Section A.2 and can be written as follows:(8)$$\begin{array}{ll} u\left(r,t\right)=\frac{2c^{2}}{R^{2}}\times & \overset{n}{\underset{k=1}{\sum }}\frac{J_{0}\left(\frac{\alpha_{0k}\times r}{R}\right)}{\Omega_{k}\times {\left(J_{0}^{\prime }\left(\alpha_{0m}\right)\right)}^{2}}\int_{0}^{t}(\int_{0}^{R}f\left(\varepsilon ,\eta \right)\times \;\varepsilon \;\times J_{0}\left(\frac{\alpha_{om}\times \varepsilon }{R}\right)d\varepsilon )\\ & \times \;\sin\left(\Omega_{k}\left(t-\eta \right)\right)d\eta . \end{array}$$

Below, three different scenarios of load on the membrane of a Winkler foundation were analyzed. They include cases of constant load and pulse load applied to the membrane center, with the addition of an external background high-frequency field. The rupture of the central link is assumed to be a fracture of an adhesive joint. The incubation time criterion is applied so as to identify the threshold load parameters necessary to cause link fracture in order to explore the dependence of the threshold values on the frequency of background vibration. It is suggested that the threshold load amplitude is the minimum of those sufficient to elicit the joint fracture.

### 3.1. Constant Load

The first case of loading considered is the combination of a concentrated constant force P combined with the background vibration field:(9)$$f\left(r,t\right)=P\left(H\left(t\right)\frac{\delta \left(r\right)}{R^{2}\times r}+\gamma \times sin\left(\nu t\right)\right),$$
where δ(r) is the Dirac delta function, H(t) is the Heaviside function, γ is a relative intensity of the background vibration field, and ν is a frequency. Thus, the substitution of this load function (9) into the form of the general solution (8) gives the temporal dependence of membrane deviation at the central point:(10)$$u\left(0,t\right)=\frac{2Pc^{2}}{R^{2}}\times \overset{n}{\underset{k=1}{\sum }}\frac{1}{\Omega_{k}\times {\left(J_{0}^{\prime }\left(\alpha_{0m}\right)\right)}^{2}}\times \left[\frac{1-\cos\left(\Omega_{k}\times t\right)}{R^{2}\times \Omega_{k}}\;+\frac{\gamma \times R^{2}\times J_{1}\left(\alpha_{0k}\right)\times w\left(\Omega_{k},\nu ,t\right)}{\alpha_{0k}}\right],$$
where $J_{1}$ is the Bessel function of the first order, and $w\left(\Omega_{k},\nu \right)$ is calculated as follows:(11)$$w\left(\Omega ,\nu ,t\right)=\{\begin{array}{l} \frac{\Omega \cdot \sin\left(t\times \nu \right)-\nu \times \;\sin\left(\Omega \times t\right)}{\Omega^{2}-\nu^{2}},\;\nu \neq \Omega \\ \frac{\sin\left(\;\nu \times t\right)}{\;\nu }-t\times \cos\left(\nu \times t\right),\;\;\;\;\;\nu =\Omega .\;\; \end{array}$$

### 3.2. Pulse Load

Another type of loading is the external pulse load, which can be represented as a concentrated force applied with limited duration *T* combined with a continuous background vibration field:(12)$$f\left(r,t\right)=P\left(\left(H\left(t\right)-H\left(t-T\right)\right)\frac{\delta \left(r\right)}{R^{2}\times r}+\gamma \times sin\left(\nu t\right)\right).$$The substitution of expression (12) into (8) allows one to obtain the solution to problems (6) and (7) for this case:(13)$$\begin{array}{ll} u\left(0,t\right)=\frac{2Pc^{2}}{R^{2}} & \times \overset{n}{\underset{k=1}{\sum }}\frac{1}{\Omega_{k}\times {\left(J_{0}^{\prime }\left(\alpha_{0m}\right)\right)}^{2}}\\ & \times \left[\frac{1-\cos\left(\Omega_{k}\times t\right)-1-\cos\left(\Omega_{k}\times \left(t-T\right)\right)}{R^{2}\times \Omega_{k}}+\frac{\gamma \times R^{2}\times J_{1}\left(\alpha_{0k}\right)\times w\left(\Omega_{k},\nu ,t\right)}{\alpha_{0k}}\right]. \end{array}$$

## 4. Beam Approximation

Similar analysis is performed to model the adhesion zone based on beam approximation, which takes into account the bending rigidity of the structure. This model is based on a Euler–Bernoulli beam on an elastic foundation under dynamic loading. The axis Ox is directed along the axis of the beam, the axis Oz is normal to the adhesive zone, and w(x,t) is vertical deflection of the beam along Oz at point x and at time t. A schematic diagram of beam and coordinate system are shown in Figure 2. Tensile forces can be ignored because their influence on the system is significantly less than that of bending forces. In this case, the adhesion zone is described by a fourth-order partial differential equation [9]:(14)$$EI\frac{\partial^{4}w}{\partial x^{4}}+kw=P\left(x,t\right)-\rho S\frac{\partial^{2}w\;}{\partial t^{2}},\;\;\;\;\;x\in \left(0,l\right),\;\;\;\;\;t>0,\;\;\;$$
where E (Pa) is the Young’s modulus, $I\;\left(m^{4}\right)$ is the moment of inertia, ρ (kg/m^3^) is the density, S is the cross-sectional area, k (Pa) is the modulus elastic foundation, P(x,t) (N/m) is the distributed external force, and *l* (m) is the beam length. The coordinate transformation and new notations are
(15)$$x^{\prime }=\frac{x}{l},\;\;\;\;\;t^{\prime }=\frac{t}{\tau },\;\;\;\;w^{\prime }=\frac{w}{l},\;\;\;\;c^{2}=\frac{\tau^{2}}{l^{4}}\frac{EI}{\rho S},\;\;\;\;\omega^{2}=\tau^{2}\frac{k}{\rho S},\;\;\;\;f\left(x,t\right)=l^{3}\frac{P\left(x,t\right)}{EI},$$
where τ is the incubation time. The equation takes the following dimensionless form (omitting the primes):(16)$$\frac{\partial^{4}w}{\partial x^{4}}+\frac{1}{c^{2}}\frac{\partial^{2}w\;}{\partial t^{2}}+\frac{\omega^{2}}{c^{2}}w=f\left(x,t\right),\;\;\;\;\;x\in \left(0,\;1\right),\;\;\;\;\;t>0,\;\;\;\;\;$$where *c* is a parameter corresponding to the wave propagation velocity, and ω characterizes the rigidity of the elastic foundation. Initial conditions corresponding to the balance position are as follows:(17)$$w\left(x,0\right)=\frac{\partial w}{\partial t}\left(x,0\right)=0.\;\;$$

Boundary conditions of three types are considered:Hinged ends of the beam:
(18)$$w\left(0,t\right)=w\left(l,t\right)=0,\;\;\;\;\;\;\frac{\partial^{2}w}{\partial x^{2}}\left(0,t\right)=\frac{\partial^{2}w}{\partial x^{2}}\left(l,t\right)=0,$$

Clamped ends of the beam:

(19)$$w\left(0,t\right)=w\left(l,t\right)=0,\;\;\;\;\;\;\;\frac{\partial w}{\partial x}\left(0,t\right)=\frac{\partial w}{\partial x}\left(l,t\right)=0,$$

Beam with free ends:

(20)$$\frac{\partial^{2}w}{\partial x^{2}}\left(0,t\right)=\frac{\partial^{2}w}{\partial x^{2}}\left(l,t\right)=0,\;\;\;\;\;\;\;\frac{\partial^{3}w}{\partial x^{3}}\left(0,t\right)=\frac{\partial^{3}w}{\partial x^{3}}\left(l,t\right)=0.$$

The solution can be obtained in a similar way to that shown above (Section A.3):(21)$$w\left(x,t\right)=c^{2}\overset{n}{\underset{k=1}{\sum }}\frac{U_{k}\left(x\right)}{\Omega_{k}||U_{k}||{}^{2}}\int_{0}^{t}\left(\int_{0}^{l}f\left(\xi ,\eta \right)U_{k}\left(\xi \right)d\xi ))\right)sin\left(\Omega_{k}\left(t-\eta \right)\right)d\eta .$$

### 4.1. Constant Load

The first case is a constant force load in the center of the adhesive zone under action of the external vibration field to be an external load:(22)$$f\left(x,t\right)=P\left(H\left(t\right)\delta \left(x-\frac{l}{2}\right)+rsin\left(\nu t\right)\right).$$For this type of loading, the solution is:(23)$$w\left(\frac{l}{2},t\right)=Pc^{2}\overset{n}{\underset{k=1}{\sum }}\left[A_{k}\left(1-\cos\left(\Omega_{k}t\right)\right)+rB_{k}g\left(\nu ,\Omega_{k},t\right)\right],$$
where $A_{k}={\left(\frac{U_{k}\left(l/2\right)}{\Omega_{k}||U_{k}||}\right)}^{2}$*,*
$B_{k}=\frac{U_{k}\left(x\right)}{\Omega_{k}||U_{k}||{}^{2}}\int_{0}^{l}U_{k}\left(\xi \right)d\xi$,
(24)$$g\left(\nu ,\Omega ,t\right)=\{\begin{array}{l} \frac{1}{\nu^{2}-\Omega^{2}}\left(\nu \sin\left(\Omega t\right)-\Omega \sin\left(\nu t\right)\right),\;\;\;\;\;\nu \neq \Omega \\ \frac{1}{2}\left(\frac{1}{\nu }\sin\left(\nu t\right)-t\cos\left(\nu t\right)\right),\;\;\;\;\;\;\;\;\;\;\;\;\;\;\nu =\Omega \end{array}\;$$

### 4.2. Pulse Load

In the second instance, another type of external load corresponding to the pulses of finite duration is also considered:(25)$$f\left(x,t\right)=P\left(\left(H\left(t\right)-H\left(t-t_{0}\right)\right)\delta \left(x-\frac{l}{2}\right)+rsin\left(\nu t\right)\right)\;,$$
where $t_{0}$ is pulse duration. For this type of loading, the solution is
(26)$$w\left(\frac{l}{2},t\right)=Pc^{2}\overset{n}{\underset{k=1}{\sum }}\left\{A_{k}\left[\left(1-\cos\left(\Omega_{k}t\right)\right)H\left(t\right)-\left(1-\cos\left(\Omega_{k}\left(t-t_{0}\right)\right)\right)H\left(t-t_{0}\right)\right]+rB_{k}g\left(\nu ,\Omega_{k},t\right)\right\}.$$

## 5. Results and Discussion

Analysis demonstrates the behavior of the limit load in dependence on various additional vibrations. For the beam approximation, Figure 3 shows the plot of the threshold values of constant load P versus the background field frequency ν at various relative intensities γ. It is clear that there are certain values of frequency of background vibrational fields that drastically decrease the threshold amplitudes of the load required for beam delamination. Consequently, an external field of relatively weak amplitude can have a significant effect on the characteristics of a plane adhesive layer fracture. Similar results with the same critical frequency were observed for membrane approximation (Figure 3d). However, a decrease in strength is not observed for all boundary conditions. In the case of free-ended beams (Figure 3c), the delamination process is not affected by the vibrational field.

Figure 4 shows the dependence of the critical amplitude P on its duration T at a certain frequency of the background field$\;\nu =\frac{\Omega_{1}+\Omega_{2}}{2}$ for the cases of the pulse load of the beam or the membrane. These plots demonstrate the decrease in the threshold amplitude with the growth of the load pulse duration, and it is almost independent of duration greater than T>0,1. This phenomenon is typical for transition to the static branch of the temporal dependencies of strength in various problems of dynamic fracture [28,29,30].

The plots in Figure 5 show the threshold amplitude of fracture pulse load dependency on the frequency ν for different values of relative intensity of the external background field γ. The first graph is plotted for a pulse duration $T_{1}=\frac{\tau }{3},\;\tau =0,2$ and the second one for $T_{2}=3\tau$, and, therefore, $T_{2}>T_{1}$.

These graphs show the existence of certain frequencies of the background field as in the membrane (Figure 5e,f) and as the beam approximation (Figure 5a–d), as has been shown above for the constant force loading. The same phenomenon of a significant decrease in the threshold amplitude is also observed, and the same influence of the external field for even weak intensity on critical load characteristics can be identified. It should be noted that only the hinged beam case has specific peculiarities, as demonstrated in Figure 5a,b. The influence of background field on the threshold amplitude of short pulse load does not completely depend on the value of γ (Figure 5a), whereas vibrations with relative magnitude γ=0.5 provide higher reduction in the critical amplitude of long pulses than those with γ=0.1 (Figure 5b). This implies that the relative energy consumption for short pulses is also less than that for long pulses since there is almost no difference in effect of weak γ=0.1 and moderate γ=0.5 background fields. A similar tendency emerges in the membrane approximation case. Figure 5e demonstrates that the effect of weak γ=0.1 background field is more obvious for short load pulses than for long pulses (Figure 5f).

This also indicates that a weak background sonic field that does not demand a large energy input is enough for a significant reduction in the threshold amplitude of a dynamic short pulse load. In order to study this phenomenon, the dependence of absolute critical amplitude of the load pulse on its duration for hinged and clamped beams is considered. A slightly different constant frequency of background vibrations is chosen from the main spectral value $\nu =1.05\Omega_{1}$ (Figure 6).

The most remarkable dependency is obtained for γ=0.1 since it demonstrates the existence of local minimum points alternating with constant phases. The global form of this dependency looks similar to that of the non-vibrational case, as the threshold amplitude becomes infinite for very short pulses and the average level of plain phases tends towards the critical value of the constant load at infinity. However, the moderate background vibrations γ=0.5 provide the main influence on fracture onset and apparently stop being “background”, because the threshold amplitude of short pulses does not increase and is still bounded when the pulse duration decays to zero. It can finally be concluded that the non-monotonic dependency of threshold amplitude on pulse duration for weak background field γ=0.1 is not trivial, and this indicates the existence of optimal regimes of impact energy input, a finding that will be the focus of our next study.

## 6. Conclusions

A number of problems of adhesive joint dynamic impact fracture in the presence of a background ultrasonic field using the Winkler approximation for the beam (membrane) on the elastic foundation were studied. Various types of boundary conditions were considered—clamped, hinged, and free ends. Threshold amplitudes of varying external load pulses dependent on ultrasonic frequency were first found in accordance with the nonlocal incubation time criterion. This enabled us to demonstrate that the threshold values of the primary pulsed load amplitude could be considerably decreased at certain frequency values of the background field. Particular calculations indicate that even a weak background sonic field is enough for a significant reduction in threshold amplitude of dynamic short pulse load. This phenomenon does not qualitatively depend on boundary conditions and bending stiffness parameters, which, on the whole, do not alter its existence. The phenomenon can be utilized in MEMS applications, with the background vibrational field as a factor controlling volatile regimes of nano-electromechanical switches.

## Figures and Tables

**Figure 1 materials-14-03188-f001:**
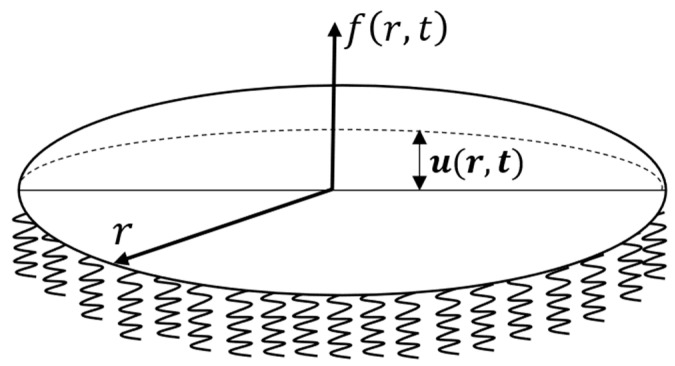
Membrane under external loading f(x, t).

**Figure 2 materials-14-03188-f002:**
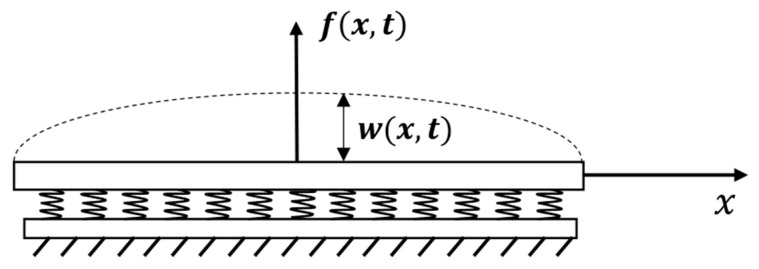
Beam under external loading f(x, t).

**Figure 3 materials-14-03188-f003:**
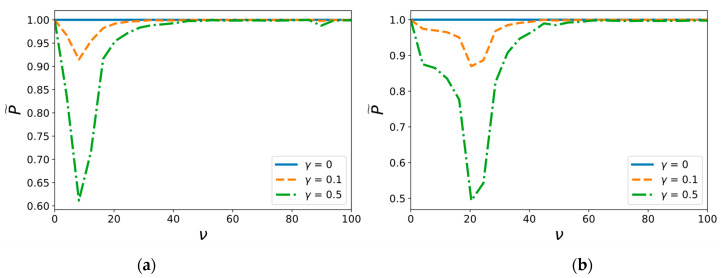
Dependence of load threshold amplitude on the background field frequency for (**a**) hinged beam, (**b**) clamped beam, (**c**) free-ended beam, and (**d**) membrane.

**Figure 4 materials-14-03188-f004:**
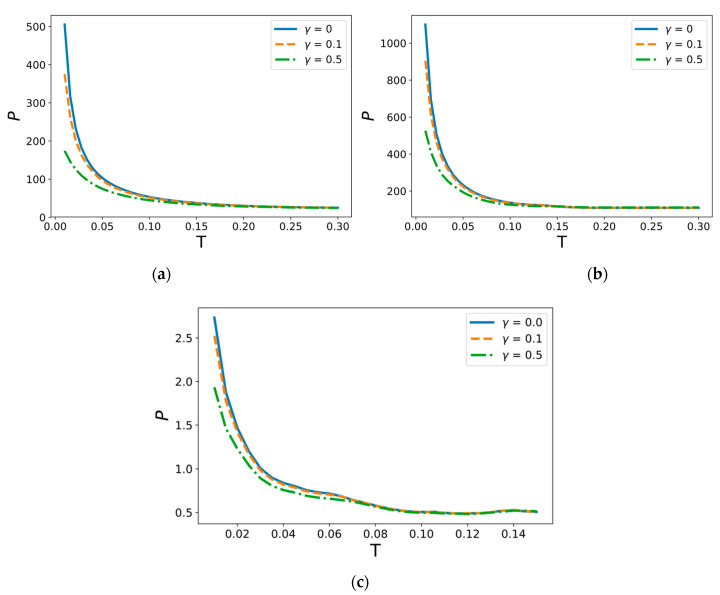
Dependence of the pulse load critical amplitude on its duration in the continuous background vibrational field for (**a**) hinged beam, (**b**) clamped beam, and (**c**) membrane.

**Figure 5 materials-14-03188-f005:**
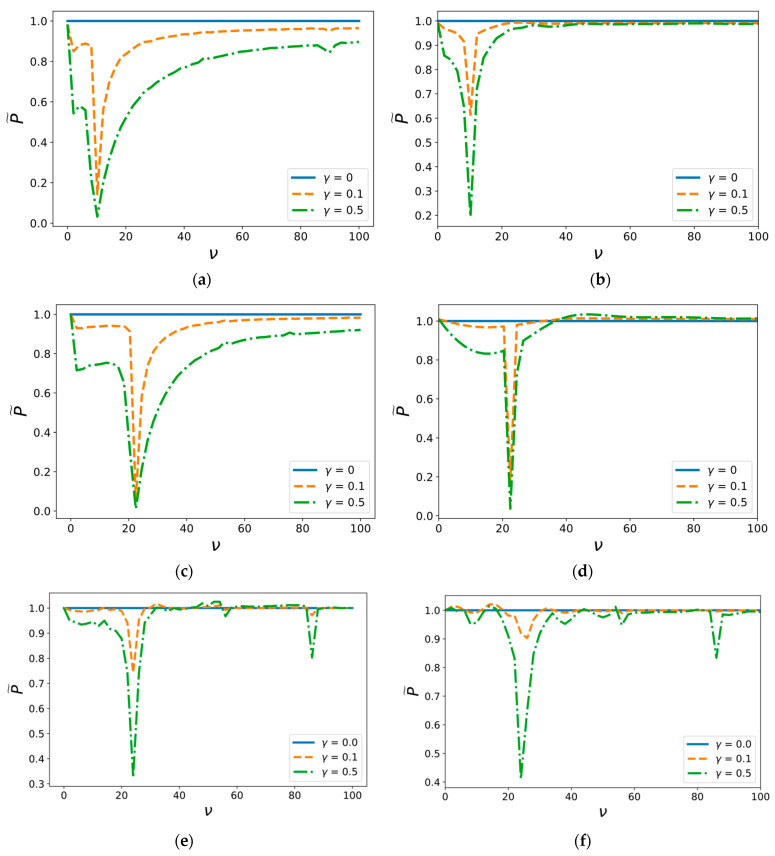
Dependence of pulse load critical amplitude on the frequency of the external background field with the pulse duration $T_{i}$ for (**a**) hinged beam, $T_{1}=\frac{\tau }{3}$ (**b**) hinged beam, $T_{2}=3\tau$, (**c**) clamped beam, $T_{1}=\frac{\tau }{3}$ (**d**) membrane, $T_{2}=3\tau$, (**e**) membrane, $T_{1}=\frac{\tau }{3}$, and (**f**) clamped beam, $T_{2}=3\tau$.

**Figure 6 materials-14-03188-f006:**
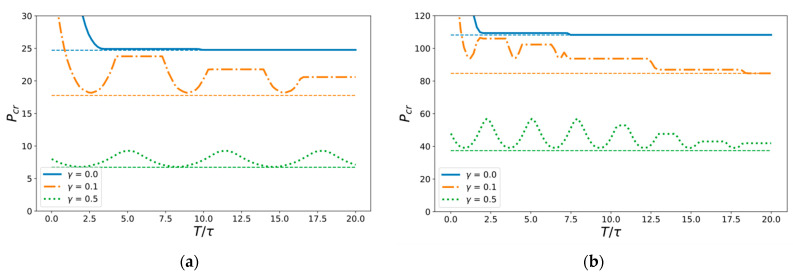
Dependence of the critical amplitude of the load pulse on its duration for (**a**) hinged and (**b**) clamped beam. Dashed horizontal lines show critical value of the constant load that corresponds to the infinite pulse duration. The external background field frequency is $\nu =1.05\Omega_{1}$.

## Data Availability

Not applicable.

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
