# Peer review of "The Influence of Background Ultrasonic Field on the Strength of Adhesive Zones under Dynamic Impact Loads"

_materials, 2021, doi:10.3390/ma14123188_

Round 1

Reviewer 1 Report

The article is very well written in terms of content. The written content deserves to be published, after addressing several comments:

  1. The content is written correctly, but completely inconsistent with the journal format. Please prepare the content according to the available template on the journal page.
  2. The authors cite too many of their own publications in the text, there are too many self-citations of their works - please take care of a more extensive review of the literature for example in the context of adhesive bonds or delamination in terms of the finite element method (DOI): 10.1016/j.compstruct.2015.06.058, 10.1016/j.compstruct.2015.11.018, 10.1016/j.compstruct.2020.113303.
  3. It is unacceptable how many papers at the end in the cited literature include the names of the authors of this paper - please expand the literature as indicated, thus broadening the horizon of the research description.
  4. Please demonstrate the novelty of this paper in relation to most similar research papers - preferably in the abstract or introduction.

Author Response

The authors are very thankful to the Reviewer for the useful comments and remarks. We took them into account in order to improve the manuscript.

  1. The content is written correctly, but completely inconsistent with the journal format. Please prepare the content according to the available template on the journal page.

We found on the Materials web cite that the free format submission is possible and we supposed that the previous version of the manuscript was proper to submit. Now we tried to format it according to the journal template.

  1. The authors cite too many of their own publications in the text, there are too many self-citations of their works - please take care of a more extensive review of the literature for example in the context of adhesive bonds or delamination in terms of the finite element method (DOI): 10.1016/j.compstruct.2015.06.058, 10.1016/j.compstruct.2015.11.018, 10.1016/j.compstruct.2020.113303.

We would like to give a special thanks to you for this particular point. The review of mentioned papers gives us the field for the further research. We also mentioned these works in the introduction section, since the similar problems are considered there. Unfortunately, the FEM simulations there do not correspond exactly to considered task in our study, but they gave us an idea how to implement our model to another problems.

  1. It is unacceptable how many papers at the end in the cited literature include the names of the authors of this paper - please expand the literature as indicated, thus broadening the horizon of the research description.

We expanded the literature and removed some links of papers with the authors names if it does not contradict to the logic of narrative.

  1. Please demonstrate the novelty of this paper in relation to most similar research papers - preferably in the abstract or introduction.

The text was rewritten in order to emphasise the novelty of the research. At first it is related to the application of the nonlocal criterion to the Winkler foundation model of the adhesive zone. This problem set allowed us to study the influence of the background vibrations on the strength of the zone. We have shown that the threshold values of the primary pulsed load amplitude can be considerably decreased at certain frequency values of the background field. We also obtained a nonmonotonic dependency of critical amplitude of the load pulse on its duration that indicates the existence of optimal regimes of loading with a minimal energy consumption to fracture. All these results are included now in the “results and discussion” section and we think that they are going to be a start point of further research.

Reviewer 2 Report

  1. The current study investigates the effect of background ultrasonic field on the performance of adhesive zoner subjected to impact loading. The authors aim to confirm whether the ultrasound wave can reduce the threshold value of impact load according to a joint model approximations and boundary conditions. The authors apply three different loadings and measure several metrics and report several findings regarding the ultrasonic wave.
  2. Please consider reviewing the abstract and highlight the novelty, major findings and conclusions.
  3. Before “In this analysis, we study the simplest one-dimensional” please answer the following question: What is the research gap did you find from the previous researchers in your field? Mention
  4. it properly. It will improve the strength of the article.
  5. Please do not use we or our, please check this issue everywhere in the manuscript
  6. Page 5-13 the authors include very detailed mathematical models, it is recommended to move these pages into an appendix and only keep the most important formulas in the main manuscript.
  7. The authors must add a list of nomenclature for all the symbols and Greek letters reported in this manuscript
  8. “However, strength decrease is not observed for all boundary conditions” could the authors explain why couldn’t be observed, please support with references if possible
  9. Combine figures 3/4/5/6 into one larger figure (recommended)
  10. “fracture is almost independent on a pulse load duration greater than ? > 0,1.” How about past studies did they find similar trends or different from yours, please discuss this further and support with references
  11. Please consider combining some figures togethers to reduce the overall number of figures
  12. Consider combining figures 7/8/9/10         
  13. The paper lacks any critical discussion
  14. The results are merely described and is limited to comparing the experimental observation. The authors are encouraged to include a discussion section and critically discuss the observations from this investigation with existing literature.

Author Response

The authors are very thankful to the Reviewer for the job and useful comments and remarks. We took them into account in order to improve the manuscript as follows:

  1. The current study investigates the effect of background ultrasonic field on the performance of adhesive zoner subjected to impact loading. The authors aim to confirm whether the ultrasound wave can reduce the threshold value of impact load according to a joint model approximations and boundary conditions. The authors apply three different loadings and measure several metrics and report several findings regarding the ultrasonic wave.
  2. Please consider reviewing the abstract and highlight the novelty, major findings and conclusions.

Our paper is pure analytical research aimed to search for new ways to control adhesive joints strength in different applications. We made some changes in the Abstract and emphasized the main goal of the study:

"... whether relatively weak ultrasound is able to decrease threshold values of the external load depending on joint model approximations such as elastic membrane or beam and various boundary conditions at the ends. The special task is to investigate the case of short pulse loading by application time-dependent fracture criterion instead of the conventional principle of critical stress."

  1. Before “In this analysis, we study the simplest one-dimensional” please answer the following question: What is the research gap did you find from the previous researchers in your field? Mention
  2. it properly. It will improve the strength of the article.

We made some changes in the Introduction and explain the choice of considered model:

“It should be noted, that generally numerical methods are not able to predict some new phenomena since they provide results for the certain values of initial parameters and do not indicate their influence to the final result. Thus, it is reasonable that the one-dimensional model of a beam and two-dimensional model of a membrane on the elastic foundation is applied for analytical study in the present research.”

  1. Please do not use we or our, please check this issue everywhere in the manuscript

The text has been corrected in an impersonal style.

  1. Page 5-13 the authors include very detailed mathematical models, it is recommended to move these pages into an appendix and only keep the most important formulas in the main manuscript.

Secondary analytical calculations were placed in the Appendix.

  1. The authors must add a list of nomenclature for all the symbols and Greek letters reported in this manuscript

The list of all symbols used in the text was added.

  1. “However, strength decrease is not observed for all boundary conditions” could the authors explain why couldn’t be observed, please support with references if possible

We just described the obtained plots and emphasised that the critical force decrease is observed in the case of free ended beam. One of the possible explanations is that the all points of the elastic system oscillate as a one whole body under background vibrations and it does not influence to the adhesive strength. The text was changed as follows:

“However, strength decrease is not observed for all boundary conditions. In case of free ended beam (Figure 3(c)) the delamination process is not affected by the vibrational field.”

  1. Combine figures 3/4/5/6 into one larger figure (recommended)

Figures 3 and 4 were combined into one and the 5-th and the 6-th were merged too. These figures show different effects, thus, combining all figures 3-6 into the one is not logical.

  1. “fracture is almost independent on a pulse load duration greater than ? > 0,1.” How about past studies did they find similar trends or different from yours, please discuss this further and support with references

This phenomenon is typical for the transition to the static branch of the temporal dependencies of strength that is indicated in the revised text and is supported by the references.

  1. Please consider combining some figures togethers to reduce the overall number of figures

We combined some figures and the overall number of figures decrease from 10 to 6

  1. Consider combining figures 7/8/9/10

Figures 7-9 were combined too. We did not add Figure 10 since it demonstrates the non-monotonic behaviour of the threshold amplitude on pulse duration, whereas Figures 7-9 show pulse critical amplitude on the frequency of the background vibrations.

  1. The paper lacks any critical discussion
  2. The results are merely described and is limited to comparing the experimental observation. The authors are encouraged to include a discussion section and critically discuss the observations from this investigation with existing literature.

The general influence of the micro-vibrational background field on adhesive joints was indicated in [7] but obviously there is a lack of proper experimental research in this field. We understand the importance of experimental verification of theoretical predictions obtained in the present study. Unfortunately, we have not found any experimental observations related to effects considered within the developed model. It inspires to plan such experimental activity in our further research.

Round 2

Reviewer 2 Report

All questions were answered. the paper can be accepted for publication

Congratulations to the authors